# Structural and Functional Studies of *S*-(2-Carboxyethyl)-L-Cysteine and *S*-(2-Carboxyethyl)-l-Cysteine Sulfoxide

**DOI:** 10.3390/molecules27165317

**Published:** 2022-08-20

**Authors:** James K. Waters, Valeri V. Mossine, Steven P. Kelley, Thomas P. Mawhinney

**Affiliations:** 1Experiment Station Chemical Laboratories, University of Missouri, Columbia, MO 65211, USA; 2Department of Biochemistry, University of Missouri, Columbia, MO 65211, USA; 3Department of Chemistry, University of Missouri, Columbia, MO 65211, USA; 4Department of Biochemistry and Child Health, University of Missouri, Columbia, MO 65211, USA

**Keywords:** amino acid stress signaling, antioxidants, carbocisteine, heavy metal cytotoxicity, green fluorescent protein, luciferase assay, NRK-52E cell line, transcriptional activation reporters, X-ray diffraction crystallography

## Abstract

Insecticidal non-proteinogenic amino acid *S*-(2-carboxyethyl)-L-cysteine (β-CEC) and its assumed metabolite, *S*-(2-carboxyethyl)-l-cysteine sulfoxide (β-CECO), are present abundantly in a number of plants of the legume family. In humans, these amino acids may occur as a result of exposure to environmental acrylonitrile or acrylamide, and due to consumption of the legumes. The β-CEC molecule is a homolog of *S*-carboxymethyl-l-cysteine (carbocisteine, CMC), a clinically employed antioxidant and mucolytic drug. We report here detailed structural data for β-CEC and β-CECO, as well as results of in vitro studies evaluating cytotoxicity and the protective potential of the amino acids in renal tubular epithelial cells (RTECs) equipped with reporters for activity of seven stress-responsive transcription factors. In RTECs, β-CEC and the sulfoxide were not acutely cytotoxic, but activated the antioxidant Nrf2 pathway. β-CEC, but not the sulfoxide, induced the amino acid stress signaling, which could be moderated by cysteine, methionine, histidine, and tryptophan. β-CEC enhanced the cytotoxic effects of arsenic, cadmium, lead, and mercury, but inhibited the cytotoxic stress induced by cisplatin, oxaliplatin, and CuO nanoparticles and acted as an antioxidant in a copper-dependent oxidative DNA degradation assay. In these experiments, the structure and activities of β-CEC closely resembled those of CMC. Our data suggest that β-CEC may act as a mild activator of the cytoprotective pathways and as a protector from platinum drugs and environmental copper cytotoxicity.

## 1. Introduction

*S*-(2-Carboxyethyl)-l-cysteine (**1**, *S*-(β-carboxyethyl)cysteine, β-CEC) is a naturally occurring, non-proteinogenic amino acid found in legumes of potential importance to economies in tropical and subtropical regions. For example, legumes of the *Calliandra* genus, which are widely cultivated in Indonesia for firewood and animal fodder, may contain up to 3% β-CEC per dry plant weight [1]. Seeds of the *Acacia* genus, which constitute a part of aboriginal diets in Australia, have been promoted as a drought-resistant food source for arid regions of Africa [2,3]. Along with **1**, these plants contain moderate amounts of *S*-(2-carboxyethyl)-L-cysteine sulfoxide (β-CECO) [3]. In humans, formation of β-CEC was established by detecting this amino acid in urine of subjects allegedly exposed to dietary or occupational acrylamide [4,5]. Scavenging of acrylamide with glutathione into allegedly non-toxic tripeptide containing **1** has been proposed as a major detoxification pathway for acrylamide in mammals [6]. In laboratory animals, β-CEC has been identified as the major adduct formed between hepatocellular proteins and acrylonitrile [7]. Cystathionine metabolites found in cystathioninuria patients include **1** [8,9].

Whereas the insecticidal potential of β-CEC has been recognized [1,10], its biological activities in mammals have not been adequately explored, with a few exceptions. Thus, β-CEC at sub-millimolar concentrations inhibited mammalian cystathionine γ-lyase [11], which is involved in endogenous H_2_S synthesis. β-CEC and other cysteine derivatives could inhibit production of atherosclerosis-promoting factors triacylglycerol and cholesterol in human hepatic cells [12]. Supplementation of experimental animal diets with **1** resulted in lower protein utilization by rats, allegedly due to decreased bioavailability of methionine [13]. In addition, β-CEC structure is a homolog of *S*-carboxymethyl-l-cysteine (carbocisteine, CMC), a clinically approved drug prescribed to patients with chronic obstructive pulmonary disease. The therapeutic activity of CMC is believed to rely on its mucolytic and antioxidant mechanisms in the airways [14,15]. The protective potential of *S*-carboxymethyl-L-cysteine sulfoxide against oxidative stress agents in airway epithelial cells has been evaluated, as well [16]. It would then be of practical interest to test the antioxidant potential of β-CEC as well.

We have recently developed a panel of insulated reporter transposons for gauging cellular signaling pathways implicated in inflammation, oxidative stress, misfolded protein responses, DNA damage, and heavy metal and xenobiotic responses [16,17,18,19]. Such reporters allow for rapid and scaled-up screening of activators or inhibitors of specific transcription factors involved in the signaling pathways, using a routine luciferase assay. This approach offers a practical platform for monitoring cytotoxicity at the level of specific cellular stress, in addition to traditional viability studies.

As a part of our studies on antioxidants capable of mitigating oxidative stress in the airway and renal cells [16,18,20], we have prepared β-CEC and its sulfoxide and report here a comparison of structure, cellular stress responses, and the cytoprotective potential of these amino acids to CMC.

## 2. Results

*S*-(2-Carboxyethyl)-L-cysteine was synthesized according to Figure 1, starting with L-cysteine and acrylic acid and following an established protocol [21]. Consequent oxidation of **1** with cold H_2_O_2_ resulted in a 1:1 mixture of two products, as evidenced from the ion-exchange chromatographic analysis (Appendix A). These were identified as *S*-(2-carboxyethyl)-l-cysteine sulfoxide diastereomers (**2** and **3**) as follows. Fractional crystallization of the mixture afforded separate crops of crystalline β-CECO which contained pure epimers **2** and **3**. The epimeric purity of the crystalline materials has been established by ion-exchange chromatography and polarimetry, and the absolute structures of the β-CECO epimers were established by the X-ray diffraction analysis data.

### 2.1. Molecular and Crystal Structures of β-CEC and Epimers of β-CECO

The molecular structures of β-CEC and its sulfoxides are depicted in Figure 1 and Figure 2 and their crystallographic parameters are given in Appendix A. A search of SciFinder and Cambridge Structural Database by both structure and chemical names revealed no previous structural description of *S*-(2-carboxyethyl)-L-cysteine or its sulfoxides, by the diffraction methods. The most closely related structures are *S*-carboxymethyl-l-cysteine [22] and *S*-carboxymethyl-l-cysteine sulfoxide [16,23], as well as several metal complexes of *S*-carboxymethyl-l-cysteine [24,25].

In all three structures, the amino acid molecules exist as zwitterions, with the deprotonated α-carboxylic group and the protonated α-amino and ε-carboxylic groups; a similar arrangement was found for close structural analogs of these molecules, such as *S*-carboxymethyl-l-cysteine (CMC) [22], (4*R*)- and (4*S*)-epimers of *S*-carboxymethyl-l-cysteine sulfoxide [(4*R*)-CMCO and (4*S*)-CMCO] [16]. There is a high level of conformational similarity between the cysteine portions in β-CEC and CMC, which feature an intramolecular hydrogen bonding between the sulfur atom S1 and the protonated α-amino group (Figure 1). There is also close resemblance in the cysteine conformations found in the molecule of (4*S*)-β-CECO [Figure 2b] and in the structure of orthorhombic (4*R*)-CMCO [16]. No such conformational similarities within this pool of molecules could be found for (4*R*)-β-CECO (Figure 2a). Its conformation is stabilized by a bifurcated hydrogen bond involving the ammonium H1B donor and two oxygen acceptors, O2 and O3, belonging, respectively, to the α-carboxylic and sulfoxide groups. For a comparison, in the triclinic (4*R*)-CMCO [16], there is also an intramolecular H-bond between the amino and sulfoxide groups, but the α-carboxylate is not involved in intramolecular interactions.

The crystal packing in β-CEC, (4*R*)-β-CECO, and (4*S*)-β-CECO is shown in Appendix A. These amino acids are heteroatom-rich, zwitterionic molecules; hence heteroatom contacts are prevailing in the crystal structures, as demonstrated in Table 1. The H ··· O contacts define an extensive hydrogen bonding network in all three crystal structures (Appendix A; Appendix A).

Molecular modeling calculations, which we have performed for the crystal structures, show that electrostatic forces arising from these contacts are the main contributors to the crystal packing energies (Figure 3, Table 2, Appendix A). Notably, there is a significant difference in total energies estimated for β-CECO epimers, with the (4*S*)-epimer having E_total_ about 100 kJ/mol higher than the value found for the (4*R*)-β-CECO crystal. Such a difference may be due to a more extensive hydrogen bonding network found in the crystal structure of **2**, as compared to that of **3** (Appendix A, Appendix A) and may explain more compact crystal packing (smaller molecular volume, Table 2) and a significantly lower solubility of the crystalline (4*R*)-β-CECO in water.

### 2.2. β-CEC Protects DNA from Copper-Dependent Oxidative Degradation

When optimizing the process of synthetic preparation of β-CECO, we observed that 30% hydrogen peroxide readily reacted with β-CEC at ambient temperatures to produce the sulfoxide, which, in turn, could further react with an excess of H_2_O_2_ to form *S*-(2-carboxyethyl)-L-cysteine sulfone. We asked then whether β-CEC could act as an efficient scavenger of peroxide at physiologically relevant concentrations in a rapid assay. When 1 mM β-CEC was incubated with 40 μM H_2_O_2_ for 30 min at room temperature, only about 10% of the peroxide was consumed (Figure 4a). For a comparison, strong antioxidants glutathione and *N*-acetylcysteine depleted 90% and 60% of the initial H_2_O_2_, respectively, while no significant reduction in hydrogen peroxide was detected in the presence of β-CECO, CMC, CMCO, or 2-aminoadipic acid, a structural analog of β-CEC and CMC lacking a thioether group.

Since the β-CEC molecule is a dicarboxylic amino acid capable of chelating transition metals, we tested the antioxidant activity of β-CEC in a role of a potential inhibitor of the copper-catalyzed Fenton reaction. Both β-CEC and β-CECO, along with its structural analogs, CMC and CMCO, protected double-stranded DNA from oxidative degradation by hydroxyl free radicals, which were produced in the copper/H_2_O_2_/ascorbate system, and this inhibitory activity was comparable to the action of glutathione, and even exceeded the activity of *N*-acetylcysteine (Figure 4b).

### 2.3. Effects of β-CEC and β-CECO on Activation of Stress and Proinflammatory Signaling Pathways in Renal Tubular Epithelial Cells (RTECs)

In order to evaluate the cytotoxicity of β-CEC in vitro, we have generated a set of reporter cell lines which release firefly luciferase in response to activation of stress-sensitive transcription factors (TFs). For this study, we have chosen rat renal proximal tubular epithelial cell line NRK-52E, an established in vitro model widely used for evaluation of potentially nephrotoxic agents [27]. General consideration of β-CEC chemical structure, a sulfur-containing dicarboxylic amino acid, suggests that this molecule could potentially interrupt with such cellular processes as proteogenesis, redox, or metal homeostasis. We have previously developed DNA constructs containing the reporter sequences that include transcription factor response element (TRE)—a binding site for specific TFs—followed by a reporter luciferase gene [19]. In particular, the sequences used in this work include the antioxidant/electrophile response element, the NF-κB binding sequence [18,19], the heat shock element, the endoplasmic reticulum (ER) stress response element, the p53 response element, and the metal response element [18]. In addition, we have assembled a novel reporter vector (Figure 5) that carries eight amino acid response elements (AAREs), which are binding sites for the transcription factors ATF2/3/4 [28]. NRK-52E cells were stably transfected with these plasmids, as well as with the reporters for transcriptional activity of NF-κB, MTF-1, Nrf2, p53, HSF-1, and ATF6. The reporter cell lines were tested for selectivity (Figure 5, Appendix A). As expected, the activity of NF-κB increased upon the treatment of the RTECs with lipopolysaccharide (LPS) and proinflammatory cytokines IL-1β and tumor necrosis factor (TNF), while Nrf2 activation was particularly sensitive to redox cycling bacterial pigment pyocyanin [18]. Respective sensors of the cytoplasmic, endoplasmic reticular, or mitochondrial proteotoxic stresses, the transcription factors HSF-1, ATF6, and ATF4, responded to a panel of stressors differentially. Among the surprising responses were the high inhibitory activity of the p53 pathway activator nutlin against ATF6, and the strong inhibition of HSF-1 and p53 with the ER stressor thapsigargin.

When NRK-52E cells were exposed to β-CEC, β-CECO, CMC, or CMCO at 0.5–4 mM concentrations for 18 h, no decrease in cell viability was detected for any of the treatments (Appendix A). However, there were cellular responses to these amino acids when activities of stress-responsive pathways were assessed in the luciferase assay (Figure 6). Specifically, activities of the amino acid stress response pathway (the transcription factors ATF2/3/4) and the electrophile/oxidative stress response pathway (the transcription factor Nrf2) have significantly and dose-dependently increased in cells treated with β-CEC, CMC, and (4*R*)-CMCO. On the other hand, none of the tested amino acids caused any increase in activities of transcriptional factors responsive to unfolded protein stress in the endoplasmic reticulum (ATF6, Figure 6) or cytosol (HSF-1, Appendix A), as well as in activities of the proinflammatory NF-κB (Figure 6) and heavy metal-sensitive MTF-1 (Appendix A), as compared to the basal levels of the TF activity. Moreover, the basal activities of NF-κB, ATF6, and HSF-1 have been suppressed by 10–40% in the reporter cells treated by all the amino acids at 0.5 mM concentrations (Appendix A).

To assess whether the amino acid stress response, which was caused by β-CEC or CMC, could be moderated by proteinogenic amino acids, we treated the ATF2/3/4 reporter cells with 4 mM β-CEC and CMC in the presence of a panel of essential amino acids, also at 4 mM concentration. The resulting data are shown in Figure 7 and Appendix A. There was a striking similarity in the ATF2/3/4 activation patterns in cells, treated with either β-CEC or CMC, in response to co-treatments with specific amino acids. Thus, *N*-acetyl-L-cysteine was the only co-treatment that decreased the ATF2/3/4 activation to the basal level. Methionine, histidine, and tryptophan significantly, but not completely, inhibited increase in the amino acid stress activation by both β-CEC and CMC, while no other amino acids could moderate the stress effect of β-CEC and CMC (Figure 7). No changes in viabilities of cells treated with the combinations were detected, with the exception of a small, 10–15%, decrease in the total transcriptional activity of cells exposed to the highest dose of *N*-acetyl-L-cysteine (Appendix A).

### 2.4. Protection of RTECs from DNA Stress Caused by DNA Intercalating Cancer Drugs

Inspired by a recent report [29] which has identified carbocisteine as a drug able to alleviate hepatocyte toxicity of oxaliplatin, we tested whether CMC and β-CEC could inhibit cytotoxicity of oxaliplatin, cisplatin, as well as other DNA cross-linking and intercalating drugs, such as anthracyclines, in the NRK-52E cell line. According to our data, β-CEC could not restore viability of the cells treated with these drugs for 24 h (Appendix A). However, as shown in Figure 8, both CMC and β-CEC, at 1 mM concentration, decreased drug-induced activation of the transcription factor p53 when co-incubated with both cisplatin and oxaliplatin. No such effect was observed for any sulfoxides of these amino acids (Figure 8) or other clinically significant DNA cross-linkers and intercalators (Appendix A).

### 2.5. Interaction of β-CEC with Environmental Pollutants

We have explored a possibility that β-CEC, as a potential dietary agent, could interact with environmental pollutants that agricultural communities are at risk of exposure to. Specifically, we considered a small panel of heavy metals (As, Cd, Cu, Hg, Pb) and nephrotoxic agents (atrazine, paraquat, diquat, ochratoxin A) known for inflicting oxidative stress. In one experiment, NRK-52E cells were exposed to varying concentrations of these pollutants in the absence and in the presence of 1 mM β-CEC for 18 h. As can be seen in the resulting Figure 9 and Appendix A, there was no effect of the β-CEC co-treatment on viability of cells exposed to the organic pollutants. On the other hand, all tested nephrotoxic metals interacted with β-CEC; while this amino acid significantly inhibited the cytotoxicity of cupric chloride, it potentiated the cytotoxicity of sodium arsenate, cadmium chloride, mercuric chloride, and lead diacetate.

In order to further explore the protective potential of β-CEC against copper cytotoxicity, we considered exposure of the cells to CuO nanoparticles, an established model of environmental copper pollutant [30]. As induction of oxidative stress is considered a major mechanism of copper cytotoxicity, we evaluated the protective antioxidant effect of β-CEC, along with β-CECO (1:1 epimeric mix), CMC, and CMCO (1:1 epimeric mix), in cells reporting activity of the antioxidant pathway regulated by the transcription factor Nrf2. As demonstrated in Figure 10, all tested amino acids offered some protection against oxidative stress induced by both soluble CuCl_2_ and insoluble CuO nanoparticles; in the case of CuCl_2_, β-CEC and CMC were somewhat more efficient antioxidants, as compared to the respective sulfoxides.

## 3. Discussion

In this study, we tested a hypothesis that a plant-derived and potentially dietary non-proteinogenic amino acid, *S*-(2-carboxyethyl)-l-cysteine, structurally and functionally resembles *S*-carboxymethyl-L-cysteine, because these compounds are homologous. β-CEC in plants is thought to be a precursor of insecticidal polysulfides [10], while CMC is a clinically utilized antioxidant and mucolytic drug [31], but can also form endogenously as a result of the addition reaction between glyoxal and cysteine [32]. Whereas structural and functional characterization of CMC is well documented [16,22,33], little is known about β-CEC, in spite of its recognized exposure to humans. Along with β-CEC, the presence of smaller quantities of its sulfoxide, β-CECO, in plant samples has been reported [3]. The structure and any other properties of β-CECO are unknown.

Similarly to the synthesis of CMCO [16], non-enzymatic mild oxidation of β-CEC by hydrogen peroxide produced a 1:1 epimeric mixture of the (4*R*)- and (4*S*)-*S*-(2-carboxyethyl)-l-cysteine sulfoxide, which could be recognized by ion-exchange chromatography and separated by fractional crystallization. It is unknown whether β-CECO determined in the plant material was represented by both of the epimers.

The X-ray diffraction analysis of β-CEC and β-CECO showed structural similarity to both CMC and CMCO, in terms of molecular conformation and charge distribution. Accordingly, the electrostatic forces acted via extensive hydrogen bonding throughout the crystals and defined the energetics of crystal packing in all crystal structures. These detailed structural data could be informative for mechanistic studies of lyase action on β-CEC, which has been suggested as a pathway to cyclic polysulfide insecticides in *Acacia* and *Albizzia* species [34].

Next, we have demonstrated that β-CEC and β-CECO, similarly to CMC and CMCO, can protect polymeric nucleic acid from copper-dependent degradation by hydroxyl free radicals. Both β-CECO and CMCO were less potent antioxidants, as compared to their respective parent amino acids. This activity is in accord with the ability of β-CEC and CMC, as well as their sulfoxides, to chelate redox-cycling catalyst Cu^2+^ and other transition metal ions [35,36] and at the same time to act as a reducing agent due to the presence of a thioether group [37]. Thus, a good copper(II) chelator but poor reducing agent, 2-aminohexanoic acid, or an excellent reducing agent but poor copper(II) chelator, *N*-acetyl-L-cysteine, did not protect DNA from copper-mediated oxidative degradation. In contrast, a strong antioxidant, glutathione, which is also a good copper binder [37], inhibited oxidative degradation of DNA most potently.

Similarities in biological activities of β-CEC and CMC were determined in the cell culture. None of these amino acids or their sulfoxides were inhibiting cellular viability in our experiments, even at millimolar concentrations that are unlikely to be achieved in vivo [38]. We have found a relatively weak increase in the transcriptional factor Nrf2 activity promoted by high concentrations of these amino acids, suggesting that none of these compounds are cytotoxic on their own, but are capable of activating the cytoprotective antioxidant pathway, in accord with a recent in vivo study on CMC boosting the Nrf2 expression in rats [39] or an in vitro study on cytoprotective effects of CMC against oxidative stress in neuroblastoma SH-SY5Y [40]. However, three amino acids, namely β-CEC, CMC, and (4*R*)-CMCO, did activate a significant response of the ATF2/3/4 reporter, suggesting either amino acid deprivation or proteotoxic stress in mitochondria [41,42]. Since other proteotoxicity sensors, such as heat shock factor-1 and the ER stress sensor ATF6, were not affected by any of the tested amino acids, it is more likely that the ATF2/3/4 activation by β-CEC and CMC proceeded via the amino acid deprivation pathway. This suggestion is in accord with an in vitro study reporting an inhibitory effect of dietary β-CEC on nutritional bioavailability of methionine in rats [13], whereas in our in vitro experiments, methionine, but also *N*-acetyl-L-cysteine, histidine, and tryptophan, could decrease both β-CEC- and CMC-induced amino acid stress in the rat renal epithelial cells, as well. To explain why, out of four sulfoxides of β-CEC and CMC, only (4*R*)-CMCO showed biological activity comparable to its parent amino acid; one may refer to our recent report [16] suggesting that this epimer, rather than (4*S*)-CMCO, can be utilized by cells via stereospecific enzymatic reduction back to CMC. For instance, in mammals, only the one epimer of methionine sulfoxide, (5*S*)-MetO, can be reduced by methionine sulfoxide reductases back to Met [43]. Incidentally, the stereochemical configurations around the sulfur atom in (4*R*)-CMCO and (5*S*)-MetO are similar, with respect to the aminocarboxylate group in these molecules. It may be suggested, then, that the inactivity of (4*S*)-CMCO and both epimers of β-CECO was due to the inability of RTECs to reduce these sulfoxides back to CMC and β-CEC.

Another functional similarity between β-CEC and CMC has been revealed when these amino acids demonstrated the ability to counteract activation of the p53, a sensor of cellular DNA damage, in RTECs by cisplatin and carboplatin. The protective effect of CMC against oxaliplatin cytotoxicity has also been reported for human hepatocytes L02 [29]. In our study, none of the sulfoxides showed the protective effect, however. In contrast, all tested amino acids showed similar activity as inhibitors of copper-promoted activation of the transcription factor Nrf2, the chief master regulator of cellular stress, in RTECs. Taking into consideration the above discussion on biological activity of (4*R*)-CMCO, this result suggests direct interactions between the metals and the amino acids. In the case of the platinum drugs, the metal prefers to bind the thioether electron donor [44], while in the case of copper, participation of the thioether or sulfoxide groups is not needed for copper chelation by the amino acids [45].

## 4. Materials and Methods

All commercial reagents and cell culture media were purchased from Fisher or Sigma-Aldrich companies.

The NRK-52E (passage 16) rat kidney proximal tubular epithelial cell line was purchased from the American Type Culture Collection. The original cells, as well as the reporter transfects, were maintained in 1:1 DMEM/F12 Ham (both from Sigma, St Louis, Missouri, MO, USA) media supplemented with 5% newborn calf serum and a penicillin/streptomycin antibiotic cocktail, 100% humidity, 5% CO_2_, and at 37 °C. The cells were subcultured in a 1:5 ratio upon reaching near confluency.

Chromatographic analysis of synthetic products was performed with help of a Hitachi 8900 amino acid analyzer (Hitachi Group, Tokyo, Japan) using a high-speed physiological column (855–4515; 6.0 mm × 40 mm) with lithium buffer system (PF1–PF5). Samples were suspended in PF1 buffer; post-column ninhydrin was used for the analyte detection at 570 nm. Mass spectra were obtained using an LTQ XL Orbitrap mass spectrometer (Thermo Scientific, Waltham, MA, USA). Optical rotation data were collected using a Jasco P-1030 polarimeter (JASCO, Tokyo, Japan). All fluorescence and luminescence measurements were taken using a Synergy MX (BioTek Instruments, Winooski, Vermont, VT, USA) plate reader.

### 4.1. Synthesis and Crystallization of S-(2-Carboxyethyl)-l-Cysteine

Synthesis of **1** was carried out following a published procedure [46]. Briefly, l-cysteine (606 g, 5 moles) and acrylic acid (396 g, 5.5 moles) in 3 L of water containing 440 g (5.5 moles) of 50% sodium hydroxide were stirred overnight at room temperature. The solution was further acidified with 315 mL (5.5 moles) of glacial acetic acid and allowed to stand at 4 °C for next 3 days. Colorless plates of chromatographically pure crystalline **1** formed during this time; the crystalline mass was filtered out, washed with cold 95% ethanol, dried in air, and used for subsequent diffraction studies and synthesis of **2** and **3** without further purification. [α]_D_^23^ −8.4° (c 1, 0.2 N HCl); lit. [47] [α]_D_^24^ −7.0° (c 1, 1 N HCl). Calc. for C_6_H_11_NO_4_S: N, 7.25%. Found: N, 7.19%. Exact mass of the [M + H]^+^ ion. Calc. for C_6_H_12_NO_4_S: *m*/*z* 194.04. Found: *m*/*z* 194.05.

### 4.2. Synthesis and Crystallization of S-(2-Carboxyethyl)-l-Cysteine Sulfoxides (**2** and **3**)

The sulfoxides were synthesized according to the method of Meese [48]. In a typical experiment, 0.1 mole of *S*-(2-carboxyethyl)-l-cysteine (19.3 g) was dissolved in 67 mL water containing 8.4 g (0.1 mole) NaHCO_3_. The solution was chilled to 4 °C and 20 mL of cold hydrogen peroxide (30%) was added. Progression of the reaction was monitored chromatographically by the amino acid analyzer. After stirring for 3 days at 4 °C, the reaction was complete. Chromatographic analysis showed formation of two diastereomers at 1:1 ratio. In some experiments, excessive H_2_O_2_ was added in order to accelerate the reaction, but this resulted in formation of the sulfone byproduct. The sulfoxides were precipitated after careful addition of 20 mL cold 5N HCl, then recrystallized from water at room temperature for 3 days, yielding crystalline product as colorless prisms containing the (4*R*)- and (4*S*)-epimers at 9:1 ratio; [α]_D_^23^ + 14.1° (c 1, 0.2 N HCl). Analysis. Calc. for C_6_H_11_NO_5_S: N, 6.63%. Found: N, 6.63%. Exact mass of the [M + H]^+^ ion. Calc. for C_6_H_12_NO_5_S: *m*/*z* 212.05. Found: *m*/*z* 212.05. An additional fractional crystallization of (4*R*)-β-CECO from water was performed at 45 °C for 18 h. Crystals containing pure (4*R*)-epimer **2** were collected by filtration, washed with cold ethanol, and dried in vacuo; [α]_D_^23^ +9.8° (c 1, 0.2 N HCl). The filtrates, containing residual amounts of the (4*R*)-epimer and all of the (4*S*)-epimer, were concentrated and subjected to additional rounds of crystallization from water, until relative content of the (4*S*)-epimer in mother liquor exceeded 90%. This solution was filtered, diluted with an equal volume of methanol, and kept at 8 °C for 2 days, to deposit monoclinic crystals of chromatographically pure (>95%) (4*S*)-epimer **3**, [α]_D_^23^ + 35.9° (c 1, 0.2 N HCl).

### 4.3. X-ray Diffraction Studies

Crystal data and experimental details of the crystallographic studies are given in Appendix A. The crystal structures were solved with the direct methods program SHELXS [49] and refined by full-matrix least squares techniques using the SHELXL suite of programs [50], with the help of Olex2 [51]. Data were corrected for Lorentz, polarization, and absorption effects. Non-hydrogen atoms were refined with anisotropic thermal parameters. The hydroxyl and ammonium hydrogen atoms were located in difference Fourier maps and were allowed to refine freely. The remaining H-atoms were placed at calculated positions and included in the refinement using a riding model. All hydrogen atom thermal parameters were constrained to ride on the carrier atoms (*U*_iso_(methine, methylene H) = 1.2 *U*_eq_ and *U*_iso_(hydroxyl, ammonium H) = 1.5 *U*_eq_). Structure visualization was carried out with the Mercury program [52].

### 4.4. Antioxidant Assays

DNA protection by potential antioxidants was evaluated by the following procedure. To a solution of 50 μg polymeric DNA (from calf thymus, Sigma) per mL of chelex-treated PBS, pH 7, were added, in order, an antioxidant, CuCl_2_ (final concentration 50 μmol/L), H_2_O_2_ (final concentration 2 mmol/L), and ascorbic acid (final concentration 2 mmol/L). The reaction was left to proceed at room temperature for 2 h and then stopped by addition of DTPA to a final concentration of 10 mmol/L. Ethidium bromide was added at 20 μg/mL, and the fluorescence of the solutions was measured at 508 nm excitation/590 nm emission wavelengths.

Peroxide-scavenging activity was evaluated using a Fluorimetric Peroxide Assay Kit (Sigma cat. #MAK165), following the manufacturer’s protocol. The assay determines residual H_2_O_2_, in the presence of an antioxidant, after a fixed time of 30 min.

### 4.5. Signaling Pathway Reporters

#### 4.5.1. Reporter Vectors

Super *piggyBac* transposase expression vector PB210PA-1 was purchased from System Biosciences (Palo Alto, Santa Clara, CA, USA). The preparation and validation of reporter plasmids carrying insulated *piggyBac* transposon constructs, which contain transcriptional response element (TRE) and reporter genes, firefly luciferase, and a copepod green fluorescent protein (cGFP), as shown in Figure 5, were reported earlier [18,19,20].

The pTR21F vector was assembled as follows: inserts containing 4 bp overhang sequences for the ligation reaction and a total of 8 amino acid response elements (AAREs) for binding the transcription factors ATF2, ATF3, and ATF4 (Table 3) were synthesized and annealed. The inserts and the larger product of pTR01F [19] digestion with the *Nhe*I/*Bgl*II were uniformly assembled into pTR21F in one ligation step. Correctness of the insertion was confirmed by DNA sequencing.

#### 4.5.2. Stable Transfections

To generate stable reporter lines, the original NRK-52E cells were seeded into wells of a 96-well plate, at 2 × 10^4^ cells per well in antibiotic-free DMEM/F12 medium supplemented with 5% NCS and left to adhere for 6 h. The cells were then treated with a mixture of 100 ng reporter plasmid and 33 ng Super *piggyBac* transposase plasmid complexed with TransIT X2 transfection reagent (Mirus) at 1:2 (μg DNA/ μL reagent) ratios. After 16 h, the regular media were added and the cells were left to proliferate for the next 48–72 h. The transfected cells were then treated with the selecting antibiotic (5 μg/mL puromycin) for another week, and the surviving cells were expanded for cryopreservation and activity validation.

#### 4.5.3. Transcriptional Activity Reporter Assay

In a typical experiment, reporter cells were seeded into wells of a 96-well plate, at 1 × 10^4^ cells per well in DMEM/F12 medium supplemented with 5% NCS and penicillin/streptomycin antibiotic and left to proliferate for 48 h. The medium was replaced with the testing medium, Corning Serum-free Medium, which is essentially a Phenol Red-free DMEM/F12 formulation containing undisclosed additions of RPMI-1640 and McCoy’s 5A, and is supplemented with 1 g/L BSA, 2 mg/L insulin, 2 mg/L transferrin, and 2 μg/L selenite. After an 18 h adaptation period, the medium was replaced with fresh testing medium, now containing stressor and inhibitor agents. By this time, the NRK-52E-based reporter cells reached near confluency and each well contained about 3.3 × 10^4^ cells. The experimental treatments lasted 18 h in standard conditions (37 °C, 100% humidity, and 5% CO_2_). The cells were then lysed in 70 μL of the luciferase reporter lysing buffer (Promega). GFP content in the lysates was determined by fluorescence in the 482(9) nm excitation/512(17) nm emission wavelength (slit width) setup; this was followed by an addition of the luciferase substrate (Promega) and luminescence readings in the wells. The GFP fluorescence values were used for both evaluation of relative cell transcriptional activity/proliferation and normalization of the reporter luciferase activities in respective wells [19].

### 4.6. Molecular Modeling and Statistical Analysis

The Hirshfeld surfaces analyses and DFT calculations of the crystal lattices, at the B3LYP/6-31G(d,p) theory level, were performed using CrystalExplorer v.17.5 software [53,54]. Statistical tests and plots were carried out by using SigmaPlot (version 11.0, Systat, Palo Alto, Santa Clara, CA, USA).

## 5. Conclusions

This study has demonstrated structural and functional similarities between natural amino acid *S*-(2-carboxyethyl)-l-cysteine and its homolog *S*-carboxymethyl-l-cysteine, a drug. Precise structures of β-CEC and two epimers of β-CEC sulfoxide and their conformational and energetic similarities to CMC and CMC sulfoxides were determined. In a DNA oxidative degradation chemical model, β-CEC and β-CECO exhibited antioxidant activities that were comparable to CMC and CMCO. In a cell-based model of nephrotoxicity, the mode of cellular stress and the cytoprotective effects of β-CEC correlated closely with those displayed by CMC, as well. Since these results were obtained in vitro only, their significance for the nutritional or therapeutic value of β-CEC could not be established. However, these data might be informative for future in vivo laboratory or clinical studies of β-CEC and could contribute methodologically to a wider use of cell signaling reporters for functional characterization of natural products, toxic agents, and pharmaceuticals.

## Data Availability

The complete crystallographic data for the structural analysis have been deposited with the Cambridge Crystallographic Data Centre, CCDC ## 1936513, 2194617, and 2194618 (www.ccdc.cam.ac.uk, accessed on 18 August 2022).

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
