# Peer review of "Structural and Functional Studies of S-(2-Carboxyethyl)-L-Cysteine and S-(2-Carboxyethyl)-l-Cysteine Sulfoxide"

_molecules, 2022, doi:10.3390/molecules27165317_

Round 1

Reviewer 1 Report

Waters et al reported the crystal structure of S-(2-carboxyethyl)-L-cysteine and S-(2-carboxyethyl)-L-cysteine sulfoxide [β-CEC, (4R)-β-CECO and (4S)-β-CECO] and performed in vitro and in vivo biochemical experiments on them. This manuscript contains comprehensive structural and biochemical results and strongly supports the target molecular function. These results will help expand researchers' knowledge in related scientific fields. I support the publication of the paper after some revisions have been made.

1.     The keywords [NRK-52E; luciferase assay; green fluorescent protein; X-ray diffraction; crystal structure] written by the author are closer to the method type than the research content. I suggest using keywords that are close to the essence of the research content.

2.     Please unify the font style in the figures. And it will be easier for the reader to see if it is centered for the letters on the left and bottom of the diagram. (e.g. Relative viability, induction fold, amino acid conc, mM, CuCl2 conc, uM)

3.     Line 523-524: Excitation and emission wavelengths must be clearly distinguished and written.

4.     ‘Copies of this information may be obtained free of charge from the Director, Cambridge 569 Crystallographic Data Centre, 12 Union Road, Cambridge, CB2 1EZ, UK. (Fax: +44-1223-336033, 570 e-mail: [email protected] or via: www.ccdc.cam.ac.uk).’ This information is not required.

Author Response

1)      The keywords [NRK-52E; luciferase assay; green fluorescent protein; X-ray diffraction; crystal structure] written by the author are closer to the method type than the research content. I suggest using keywords that are close to the essence of the research content.

Response: Per Reviewer’s suggestion, we have expanded the keywords list.

2)      Please unify the font style in the figures. And it will be easier for the reader to see if it is centered for the letters on the left and bottom of the diagram. (e.g. Relative viability, induction fold, amino acid conc, mM, CuCl2 conc, uM)

Response: We have unified the font style in Figure 1 and centered the axes legends in Figures 6-10

3)     Line 523-524: Excitation and emission wavelengths must be clearly distinguished and written.

Response: Corrected

4)   ‘Copies of this information may be obtained free of charge from the Director, Cambridge 569 Crystallographic Data Centre, 12 Union Road, Cambridge, CB2 1EZ, UK. (Fax: +44-1223-336033, 570 e-mail: [email protected] or via: www.ccdc.cam.ac.uk).’ This information is not required.

Response: This sentence has been removed from the manuscript, per suggestion.

Reviewer 2 Report

The article titled  Structural and Functional Studies of 2 S-(2-carboxyethyl)-L-cysteine and S-(2-carboxyethyl)-L-cysteine 3 Sulfoxide. after consideration of the following comments.

.

1)      Abstract, conclusion and results are not clear.

2)      The rational for this study should be improved

3)      Methods , authors grams and moles so they should be mentioned only grams

4)      Methods, authors should record melting and biologging point for synthesized compounds.

5)      NMR spectra of compounds should be added.

6)      Conclusion should be added as separate section not involved through manuscript (line 393).

7)      References should be updated as only three references (2021-2022)..

Author Response

1)      Abstract, conclusion and results are not clear. 

Response: This comment is too vague to be of any use to us in improving the manuscript. The Reviewer raises no specific questions that are unanswered by our manuscript. While we certainly thank this Reviewer for taking time to examine our manuscript and offer feedback, in our opinion most of the Reviewer’s comments are simply opinions on what the content of a manuscript should be. We feel that we have met the publication standards for reporting a new compound in Molecules (or most peer-reviewed journals for that matter). The reviewer has offered no critical feedback that challenges our interpretations of the results or that, in our opinion, would significantly improve the manuscript if we incorporated it.

2)      The rational for this study should be improved 

Response: In our opinion the rationale is clearly stated multiple times throughout the manuscript. The reviewer does not state what he/she believes to be our rationale in his/her own words.

3)      Methods , authors grams and moles so they should be mentioned only grams 

Response: The use of moles in synthetic chemistry is generally preferred as it allows chemists to easily change the scale of experimental procedures and understand the reaction at a molecular level.

4)      Methods, authors should record melting and biologging point for synthesized compounds. 

Response: The Reviewer gives no justification for these additional experiments. Melting points are not commonly measured, especially for thermally unstable compounds, such as cysteine derivatives. Reference #47 provides only an interval in which CMC and β-CEC decompose.

We are unsure of what biologging is, in the Reviewer's interpretation. From what we can find it refers to the technique in biology of using radio tags to log the movements of animals. However, our experiments employed cultured rat cells, not rat animals.

5)      NMR spectra of compounds should be added. 

Response: The structures are identified unambiguously by SCXRD, mass-spectrometry, polarimetry, and chromatography. We do not see the necessity for additional chemical characterization, and the reviewer raises no arguments for why such additional experiments are necessary.

6)      Conclusion should be added as separate section not involved through manuscript (line 393). 

Response: Per request, we have added the Conclusions section in the manuscript

7)      References should be updated as only three references (2021-2022).

Response: While there are no other recent publications on β-CEC besides those that we have referenced in the manuscript, CMC is much more widely studied. Per Reviewer’s request, we added 3 most recent references (##38-40), covering CMC, to the Discussion section.